# Accurate Detection of Out of Body Segments in Surgical Video using Semi-Supervised Learning

**Maya Zohar**[1]                                             MAYA@THEATOR.IO
**Omri Bar**[1]                                                 OMRI@THEATOR.IO
**Daniel Neimark**[1]                                        DANIELN@THEATOR.IO
**Gregory D. Hager**[2,1]                               HAGER@CS.JHU.EDU
**Dotan Asselmann**[1]                                  DOTAN@THEATOR.IO

[1] *theator Inc., San Mateo, CA, USA.*

[2] *Department of Computer Science, Johns Hopkins University, Baltimore, USA.*

## Abstract

Large labeled datasets are an important precondition for deep learning models to achieve state-of-the-art results in computer vision tasks. In the medical imaging domain, privacy concerns have limited the rate of adoption of artificial intelligence methodologies into clinical practice. To alleviate such concerns, and increase comfort levels while sharing and storing surgical video data, we propose a high accuracy method for rapid removal and anonymization of out-of-body and non-relevant surgery segments. Training a deep model to detect out-of-body and non-relevant segments in surgical videos requires suitable labeling. Since annotating surgical videos with per-second relevancy labeling is a tedious task, our proposed framework initiates the learning process from a weakly labeled noisy dataset and iteratively applies Semi-Supervised Learning (SSL) to re-annotate the training data samples. Evaluating our model, on an independent test set, shows a mean detection accuracy of above 97% after several training-annotating iterations. Since our final goal is achieving out-of-body segments detection for anonymization, we evaluate our ability to detect these segments at a high demanding recall of 97%, which leads to a precision of 83.5%. We believe this approach can be applied to similar related medical problems, in which only a coarse set of relevancy labels exists, currently limiting the possibility for supervision training.

**Keywords:** Surgical Intelligence, Semi-Supervised Learning, Deep Learning, Surgical Video Anonymization, Out of Body Detection.

## 1. Introduction

It has been estimated that 312.9 million surgical procedures were performed worldwide in 2012 (Weiser et al., 2016). Estimations also show that 9 million procedures will encounter major complications (Maier-Hein et al., 2017). A big portion of these procedures is conducted using a minimally invasive surgery (MIS) approach. For example, in the laparoscopic approach, the surgeon performs the operation with the aid of a video feed from a laparoscopic camera. Combining video analysis abilities with such high volume data can revolutionize the surgical domain and ultimately improve patient care.

Recent development of cloud and on-premise big data storage warehouses, and high-quality recording hardware, is leading surgical departments to save and store their performed

procedures in video archives. Combining unprocessed surgical videos with the digitization of electronic patient records produces valuable large-scale databases. Properly organized video archives, that enable easy indexing and quick video navigation, can be utilized to train medical personnel through a debriefing process, thereby improving patient care.

As surgical videos are recorded "in the wild", they often contain long non-relevant segments. These segments can be divided into two main groups. (1) Non-relevant segments occurring before the actual start of the procedure or after the actual ending of the procedure. And (2) intraoperative non-relevant segments, for instance when the camera is pulled out for cleaning, or in case of a corrupted, black or very dark video segments. The length of these segments can vary between a few seconds to tens of minutes. As no surgical related information exists within these segments, once detected they can be discarded and removed from the video before storing it. Thus, reducing storage size needed for saving the video content.

Importantly, detecting non-relevant segments is also crucial for maintaining staff and patient privacy. As videos might contain medical personnel or patient identified footage, it is essential to identify these segments and either blur or remove them altogether from the video. Automatic anonymization capabilities will allow medical centers to work and share their databases more openly.

Surgical intelligence is a growing field in ML-based medical applications (Maier-Hein et al., 2017). As models are trained on surgical video datasets they should handle such non-relevant segments during training. For an ML-based system, learning from these noisy samples influences the learning process and impacts the ability to learn actual surgical context. By removing noisy segments one can expect models to learn a better representation which eventually leads to enhanced performance (Twinanda et al., 2014a).

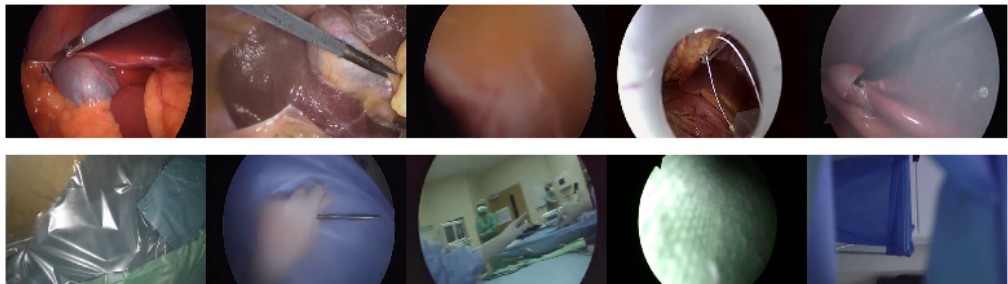

Figure 1: Examples of surgical-related samples (top row) and non-relevant samples (bottom row).

The work of (Münzer et al., 2013) approached this problem by defining three irrelevance classes: dark frame, out-of-patient frames, and blurry frames. Their proposed method extracts visual color-based features (such as hue histograms) which act as indicators for each class existence, these indicators are used to train and classify input frames. In (Atasoy et al., 2011), a low dimensional representation, based on image frequency domain energy values, was adopted to cluster uninformative frames and patient-specific segments in Gastro-

intestinal (GI) endoscopy. However, both of these methods are based on supervised learning and require labeling the non-relevant segments. The work of (Twinanda et al., 2014b) proposes to avoid the need for labeling by an unsupervised method that operates on an RGB histogram but requires manually setting a threshold empirically after observing a few videos from the dataset. We provide a comparison of our method results and those of (Twinanda et al., 2014b) in Appendix B.

Although these studies show promising results, they are not sufficient for production-level relevancy identification. They depend on highly engineered features and use specific dataset properties, thus their ability to generalize to new unseen laparoscopic videos, in an unknown recording environment, remains questionable. Our proposed method utilizes deep learning holistic representation of an image that yields a robust and generalized model able to handle surgical videos recorded in an "in the wild" setting.

Such representation is able to handle a variety of edge cases that exist in surgical videos. For example, a video stream might be acquired with a circular or part-circular black region surrounding the actual region of interest (Figure 1). Instead of dealing with these variations using cherry-picked features, our model is able to learn all variations within our dataset and adjust to various types of recording hardware and camera manufacturers.

The goal of this work is training a deep learning model capable of accurately detecting non-relevant segments throughout an entire surgical video. We process each video second as an independent sample and consider non-relevant samples as the positive class and surgery-related samples as the negative class – forming a binary classification problem.

The main challenge of this type of training is the need to correctly annotate videos with relevant or non-relevant segments. Manually labeling every second of a procedure video with such labels is a cumbersome and exhausting process. Non-relevant segments are sparse and appear sporadically throughout a video. Therefore, achieving enough labeled data in order to train Deep Convolutional Neural Networks (DCNNs) is a lengthy and resource-intense process. In contrast, solely indicating the actual start and end of a procedure is a relatively quick and easy task. Human annotators can review a video at high speed and locate the transitions between non-relevant early-start or late-end of a procedure recording. Labeling the start-end task leads to a weakly labeled noisy dataset, consists of a large number of positive segments and a noisy set of negative segments which include positive seconds falsely labeled as negative. Our method utilizes this noisy set of labeled data as the basis for training a model for the full task of non-relevant detection. This is done by iteratively training a model with incomplete supervision. First, on the initial set of non-complete labels, and then on a new set of labels achieved by using the resulting model. This iterative process is then repeated in order to achieve optimal performance.

## 2. Methods

### 2.1. Datasets

Our dataset contains 640 videos of laparoscopic cholecystectomy (minimally invasive gallbladder removal), curated from six different medical centers. We first randomly select 20 videos for the validation set and 20 videos as a test set. To explore different training setups, in which (1) the same train set and (2) a different train set of 100 videos are used for training, we randomly divided the remaining 600 videos into six train subsets, from which

Table 1: Number of positive samples, negative samples and unique non-relevant segments in the train, validation and test sets.

| Set | Surgery-related | Non-relevant | Non-relevant segments |
|---|---|---|---|
| Train | 180767 | 23499 | - |
| Validation | 35733 | 3320 | 64 |
| Test | 43083 | 5221 | 79 |

one subset is defined as the initial baseline train set in our experiments. Since each second of a surgical video is considered as a single sample, this translates to 204,266 samples in the baseline train set and 39,053 and 48,304 samples in the validation and test set, respectively (Table 1).

Train and evaluation (validation and test) sets were manually annotated using different labeling definitions. For the train set, the annotator task was to locate and label the actual start- and end-time of the surgery. Therefore, all seconds before the start-point and after the end-point, while the camera is outside the patient's body, are annotated as non-relevant (positive) segments. Video seconds that normally would be considered as non-relevant segments during the time of surgery are ignored and falsely labeled as surgery-related (negative) samples at this point. In contrast, the validation and test sets were manually annotated with full coverage of non-relevant labeling throughout the entire surgical video (Table 1).

Figure 2.B shows the number of training samples per class for each training iteration. The initial baseline train set contains only 23499 positive samples and is skewed toward the negative class which has 180767 samples. In contrast, the final iteration contains 33871 positive samples and 170395 negative samples, which is a 44% increase in positive samples.

Non-relevant segments are not a common occurrence in surgery. When these do occur, for example when the camera is extracted so it can be wiped clean, most last for only a few seconds. Thus, the number of non-relevant samples is significantly smaller than the number of surgery-related samples, and the classes are not represented equally during training. A classifier trained on an imbalanced dataset is more likely to be biased towards the majority class and show poor results on the minority class, which in our case is the class of interest. To tackle this problem we applied a custom batch scheduling method and forced a balance sampling in each mini-batch (Buda et al., 2018). We chose to deal with this problem on the data level and use a straightforward approach of oversampling the minority class while constructing the mini-batches during training. This is done by changing the class distribution and simply replicating randomly selected samples from the minority class. We use several augmentations during training to limit overfitting that might be caused by such random minority oversampling.

### 2.2. Implementation details

The videos are pre-processed using FFmpeg 3.4.6 on Ubuntu 18.04 and all video streams are encoded with libx264, using 25 frames per second (FPS). The video width is scaled to 480 and the height is determined to maintain the aspect ratio of the original input video.

The model architecture is ResNet-18 (He et al., 2016), and weights are initialized to a pre-trained ImageNet network (Deng et al., 2009). We use transfer learning and finetune the model by replacing the last layer output to a two-class classification layer and training the last layer only on the surgical video dataset (Girshick et al., 2014; Goodfellow et al., 2016). The model was trained to classify each independent second as a surgery-related or non-relevant sample. During training, we use different augmentations on each sample; random rotation in a range of ±5 degrees, random resize with a scale of 0.96-1 and aspect ratio of range 0.95-1.05, random horizontal flip and random crop to a size of $224 \times 224$. During model evaluation, each sample is resized to 300 (on its smaller edge) and then center cropped to a size of $224 \times 224$. All samples are normalized based on ImageNet mean and standard deviation values (mean = [0.485, 0.456, 0.406], std = [0.229, 0.224, 0.225]).

The model is trained using a Stochastic Gradient Descent (SGD) optimizer, with an initial learning rate of 0.001 and a momentum of 0.9. We use a cross-entropy loss function and train for 20 epochs using a mini-batch of 64 samples. We gradually decrease the learning rate during training by a factor of 0.1 after 10 epochs and again after 15 epochs.

Our method is based on semi-supervised iterative learning. The first model was trained on a weakly labeled dataset, meaning, only part of the video is annotated correctly. Then, we use the resulting model to predict the train set samples and modify its labels. The new set of labels is then used again to train a new model. We repeat this process six times for every experiment setup.

Since predictions are produced for each video second, the model might classify a single second as non-relevant within a large surgical-related segment. In a typical surgical procedure some continuity is expected, and it is unlikely that the camera is outside the patient's body for such a short time. To avoid this type of error and smooth the results we apply a smoothing filter on the final output predictions. We used morphological opening and closing operations with a kernel equal to three to filter the predictions vector. Applying this filter improves the results by about 1%, and in some cases might remove a true detection of a non-relevant samples. However, in terms of video sequential context it contributes to maintaining smooth continuous predictions.

## 3. Results

To evaluate the performance of our proposed method we report an independent accuracy for each class and its mean accuracy value. Non-relevant accuracy is representing the method's recall value. Since our main goal is detecting the minority non-relevant class with a high recall we also report the corresponding precision value. As the recall represents the percentage of true positive samples correctly classified as positive samples, the precision value complements its information by showing the percentage of samples classified as positive which are in fact true positive samples.

We explore three different setups, studying the impact on performance when training with a constant or non-constant train set and in case model weights are randomly initialized or loaded from a pre-trained model.

### 3.1. Using a constant train set in each iteration

In the first experiment, we train a model by using only the baseline train set. Training starts from the weakly labels set and the resulting model is used to predict the train set videos and re-label them. Then a new training cycle initiates on the modified labels. The train-annotate process is repeated six times.

Figure 2 shows how accuracy varies across different iterations as the train set labels are modified by the previous model. The number of non-relevant training samples is growing as iterations progress, demonstrating the effect of the re-labeling process. The initial non-relevant detection accuracy after the first iteration is 86.3% while the final iteration model reaches 96.4% accuracy on the validation set. Similar results are achieved on the test set, where the final iteration model reaches 96.9% accuracy in detecting the non-relevant segments and 92% in detecting the surgery-related segments.

Applying the post-processing filter to the last iteration predictions leads to a non-relevant accuracy of 96.6% but an improved surgery-related accuracy of 94%. At this operating point of a 96.6% recall, our model achieves 66.2% precision.

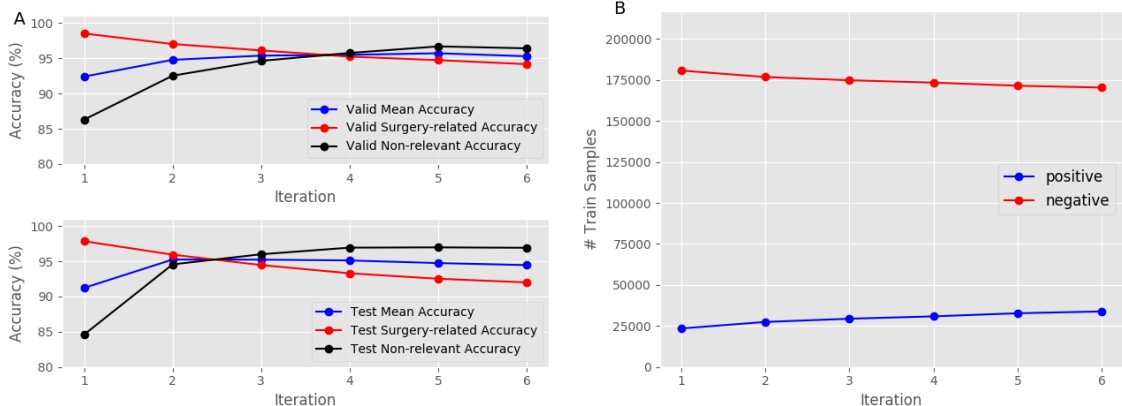

Figure 2: Finetune an ImageNet ResNet-18 model on the baseline train set. Iteratively updating the baseline train set labels using the model of the previous iteration. **A.** Results for the validation (top-left) and test (bottom-left) sets. The red and black lines show the accuracy of the surgery-related and non-relevant class, respectively. The blue line is the average of both accuracies in each iteration. **B.** Number of training samples per iteration as a result of the re-label step.

### 3.2. Using a different train set in each iteration

Now, instead of using the same train set in each iteration, which might be biased and prone to overfitting the samples the model was trained on, we update the train set in each iteration. This is done by replacing the train set with a new set of 100 videos and using the previous iteration model in order to label the new set samples.

Figure 3 shows the progress of each class accuracy for every iteration. High non-relevant accuracy of 98.8% and surgery-related accuracy of 96.5% are achieved on the validation

set after two training iterations. This increase in performance compared to the previous experiment (3.1) is due to the new accurate train set samples. The final non-relevant accuracy reaches a near-perfect result of 99.85%, but with a low surgery-related accuracy of 90% compared to the two iterations model.

The best model is achieved after two iterations and produces the highest accuracy on average for both classes. On the test set, this model shows similar results of 97.5% non-relevant accuracy and 96.2% surgery-related accuracy. Applying the post-processing filter on the predictions of this model yields an improved performance of 97% for the non-relevant accuracy and 97.7% for the surgery-related accuracy. At this operating point with a recall of 97%, the model achieves 83.5% precision.

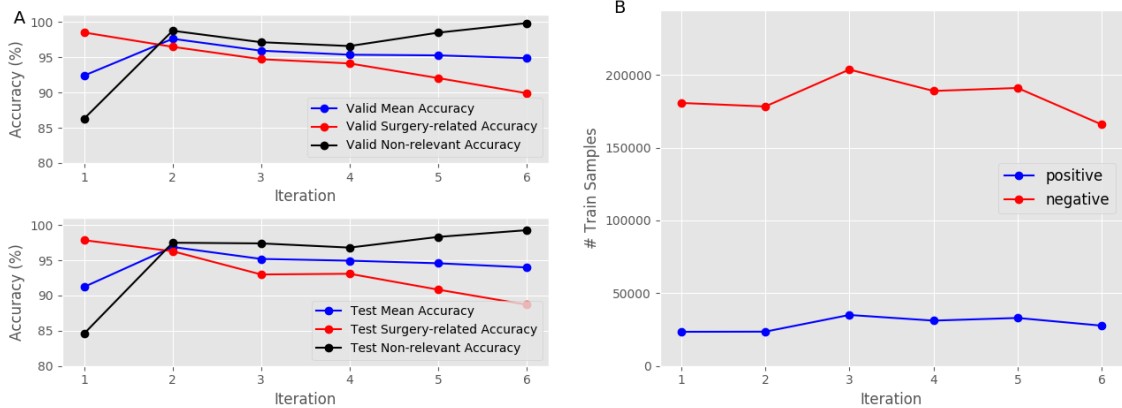

Figure 3: Finetune an ImageNet ResNet-18 model on different train sets. Iteratively replacing the train set samples and updating the labels using the model of the previous iteration. **A.** Results for the validation (top-left) and test (bottom-left) sets. The red and black lines show the accuracy of the surgery-related and non-relevant class, respectively. The blue line is the average of both accuracies in each iteration. **B.** Number of training samples per iteration as a result of the re-label step.

### 3.3. Using a different train set and fine-tuning from a previous iteration model

In the final experiment, in addition to updating the train set and using the previous iteration model to label its samples, we also initialize the next iteration model with the weights of the previous iteration best model. In practice, this means the classification layer is initialized using a previously trained model instead of a random initialization as done before. Now, in every iteration the starting point should be improved, both in terms of better labeling for the training samples and also in terms of a pre-trained classification layer.

After two iterations, the model achieves a non-relevant accuracy of 98.7% and 96.5% surgery-related accuracy on the validation set (Figure 4). At the final iteration, the non-relevant accuracy is close to perfect with 99.3% however, the surgery-related accuracy drops to 91.8%.

On the test set, the best model from the second iteration archives 97.2% non-relevant accuracy and 96.3% surgery-related accuracy. Applying the post-processing filter on this model predictions yield a non-relevant accuracy of 96.6% and 97.7% surgery-related accuracy. Similar to the setup of 3.2, at this operating point with a recall of 96.6%, our model achieves 83.5% precision.

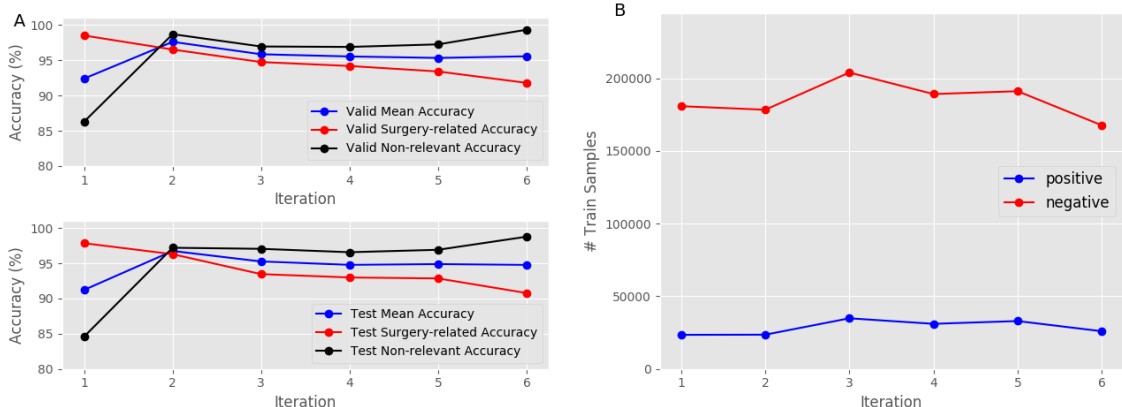

Figure 4: Finetune a ResNet-18 model on different train sets. Iteratively changing both the train set samples and labels using the model of the previous iteration and also the initial model weights to the previous iteration model. **A.** Results for the validation (top-left) and test (bottom-left) sets. The red and black lines show the accuracy of the surgery-related and non-relevant class, respectively. The blue line is the average of both accuracies in each iteration. **B.** Number of training samples per iteration as a result of the re-label step.

### 3.4. Error analysis

The main classification errors are surgery-related samples misclassified as non-relevant. Reviewing these segments shows that such errors are mostly related to blurred or unusual intraoperative segments, which share similar characteristics with segments captured outside the patient's body. Figure 5 demonstrates the model predictions for a few successful and misclassified samples.

A different type of error is related to the annotation definition in the transition segments, when the camera is entered or pulled out of the patient's body. These segments could be classified either as relevant or non-relevant in an inconsistent manner. Figure 6 shows a few examples of this transition. In some cases, the inner organs are still visible, but in others, the trocar presence is more significant. Therefore, different annotators labeled similar segments differently. For our end goal of anonymization, errors in segments that are part of the transition phase can be considered as "don't care" segments and have a minor effect on the final anonymized video.

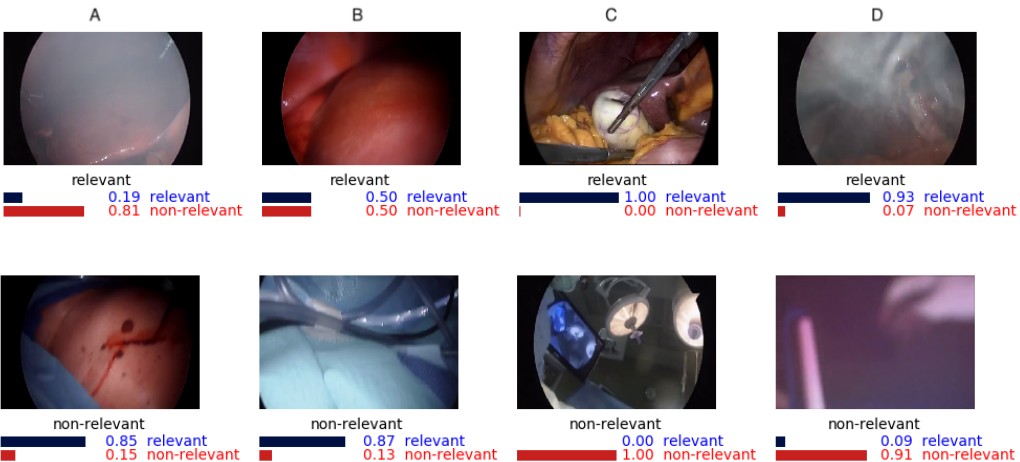

Figure 5: Surgical-related (top row) and non-relevant (bottom row) samples. The centered text below each image indicates the true label and the color bars represent the model prediction scores. Column **A** demonstrates a reasonable misclassification due to image characteristics (unusual colors and blurring). Column **B** shows unexpected misclassification samples. Column **C** presents typical class samples with accurate predictions. Column **D** demonstrates high accurate predictions for unusual blurring frames.

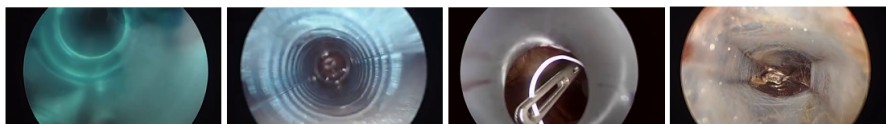

Figure 6: Examples of transition phase frames which represent similar surgery context, when the camera is taken in or out of the patient's body. The first image on the left demonstrates the characteristics of a non-relevant frame, while the others are more ambiguous. In the two images on the right, inner organs are visible, and the colors are more similar to surgery-related segments.

To further illustrate how the iterations approach impacts the model performance we show a comparison of each iteration predictions and the true labels on a single surgical video (Figure 7). In addition, in order to show the effect of filtering the predictions, we also show a comparison of model predictions before and after applying the filter (Figure 8).

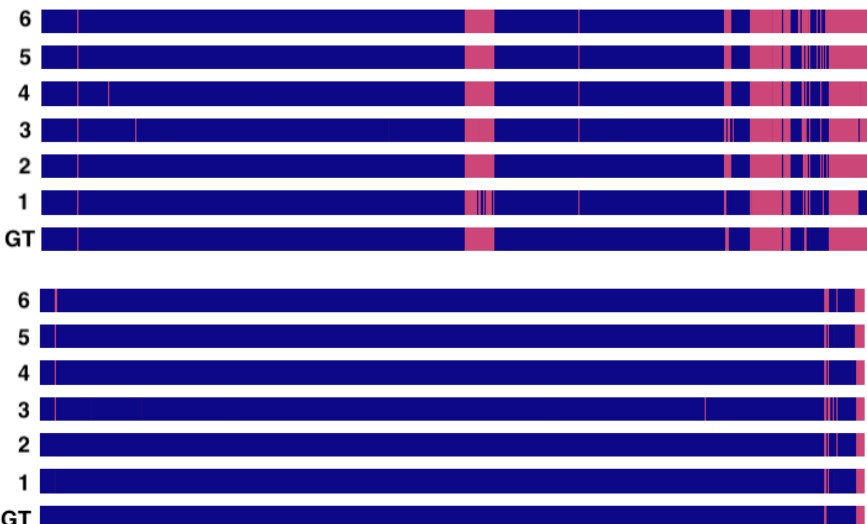

Figure 7: Color illustration of model predictions at each training iteration (top rows) vs. the ground truth labels (bottom row). The surgery-related class (blue) and non-relevant class (pink) predictions are depicted for each iteration. Each block represents a single video.

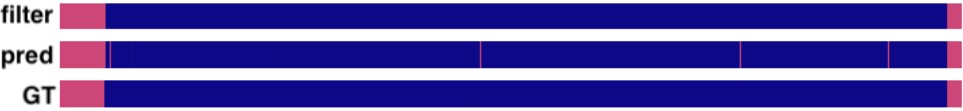

Figure 8: Color illustration of model predictions and true labels for the surgery-related class (blue) and non-relevant class (pink) before and after applying the filter.

## 4. Conclusion

In this work, we proposed a framework based on semi-supervised learning to iteratively train a model for non-relevant segments detection in surgical videos.

Given enough labeled data, solving the non-relevant segments detection problem is a relatively easy task. The main challenge that remains is handling the missing labeled data to support such supervised learning process. Our solution aims to tackle this issue by using labels from a similar subtask and iteratively learn the labels needed to solve the non-relevant segments detection task. The imbalance characteristics of our train set require a pre-processing step, in which we oversample the minority class in order to balance the distribution of samples during training. To maintain the continuous nature of the video and reduce noisy false predictions we propose a simple filter to smooth the results at a final post-processing step.

Comparing the results of 3.2 and 3.3 shows no improvement when using the previous model weights to initialize the model. This implies that the benefit of initializing the classification layer from a model trained on our specific problem versus a random initialization is not significant. In addition, comparing the results of 3.1 and 3.2, a major improvement is achieved as a result of adding new training samples between each iteration.

Although less complex models could yield relatively good results for this task, handling surgical videos anonymization for privacy purposes demands a high level of certainty, which these models lack. The novelty of our model is its abilty to discern between surgical-related and non-relevant segments with a mean class accuracy higher than 97% and a high recall of 97% at a precision level of 83.5%.

Semi-supervised learning is a promising approach that can lead to high-level performance in surgical video analysis. Identifying specific data properties, that efficiently generate even weakly labeled datasets, opens the door to applying the approach described in this work and gradually shifting from weak class labeling toward a clean set of labels. Since this property exists in other imaging datasets, the impact of our findings is not limited and could be used in other surgical, interventional and medical fields in which similar tasks can prove advantageous. These potentially include not only endoscopy-based surgical videos but other procedures such as cardiac catheterization, urologic and gynecologic catheterizations and a wide range of minimally invasive approaches to patients. Regardless of the domain, for video analysis to become standard of care in the clinical setting, methods such as the one we propose are essential in order to glean valuable insights while protecting the privacy of both patients and staff.

## Acknowledgments

We would like to thank Ross Girshick for many helpful discussions and valuable comments.

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

**Appendix A.**

To extend the error analysis (3.4) and better understand the type of errors, we examined intraoperative frames classified as surgical-related at the first few iterations and misclassified as non-relevant at later iterations.

Apparently, most of these frames are segments captured during surgery in which the camera is in a static position and there is no actual visible surgical activity. Such frames are either extremely blurry or very bright (Figure 9).

Since the initial training set contains out-of-body segments that share similar characteristics, the model learns those segments as non-relevant and tends to recognize similar surgical-related segments falsely as non-relevant. However, for the end goal of anonymization, these segments are uninformative and could be considered as "don't care" segments. Figure 9 demonstrates the change in model confidence for a single image throughout the training iterations.

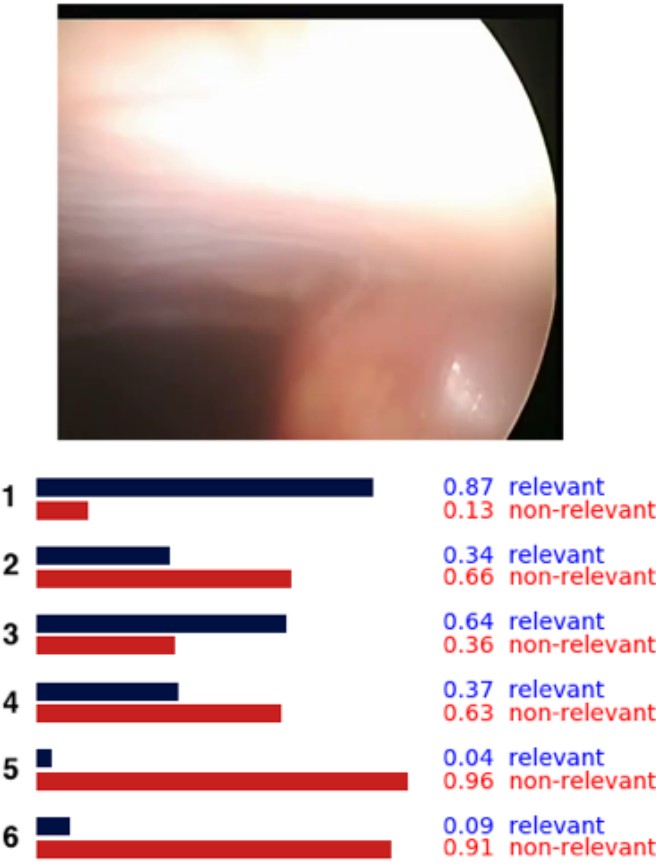

Figure 9: The model SoftMax scores at every iteration for a single surgical-related frame.

## Appendix B.

In order to achieve a comparison to (Twinanda et al., 2014b) approach, we followed their method description and calculated RGB histograms for our train set. These histograms are then used to select the best thresholds for classification. Applying the thresholds on the validation and test sets yields significantly lower results compared to our method (Table 2).

Table 2: Our method results compared to Twinanda et al. approach. Reported on the validation / test sets.

|  | Mean unweighted accuracy | Precision | Recall |
|---|---|---|---|
| Twinanda et al. | 74.2% / 70.4% | 28.6% / 30.5% | 63% / 56.4% |
| Ours | **97.8% / 97.35%** | **78% / 83.5%** | **98.2% / 97%** |

Our dataset was curated from several different medical centers, and thus span a variety of recording systems. In order to better understand how does our method generalize for different medical centers, we calculate the mean unweighted accuracy of each medical center separately in the validation and test set, and the corresponding mean and std values. The results in Table 3 demonstrate that our method generalize well across different medical centers, and is significantly better than the results of Twinanda et al.

Table 3: Evaluating the generalization for different medical centers. The results are the mean and std values of the mean unweighted accuracy, calculated separately for each medical center.

|  | Validation | Test |
|---|---|---|
| Twinanda et al. | $0.676 \pm 0.096$ | $0.685 \pm 0.0712$ |
| Ours | $0.975 \pm 0.0146$ | $0.955 \pm 0.0283$ |

