# OpenReview forum: "Accurate Detection of Out of Body Segments in Surgical Video using Semi-Supervised Learning"
_MIDL.io/2020/Conference — MIDL 2020_

### Official Review · AnonReviewer1 · 2020-02-23
**A Res-Net classifier for laparoscopic videos**

**Rating:** 2
**Confidence:** 5

**Summary:**

The presented paper aims to label and remove irrelevant sequences from laparoscopic videos. This is done with manual labelling and a ResNet-18.
Motivation is based on anonymisation and data cleansing. Iterative refinement is claimed to be semi-supervised learning. Several experiments are proposed and results are presented.



**Strengths:**

- automatic patient data anonymity and data cleansing are important topics
- the results look good with a big but (see below)
- this is clearly an application paper, testing well known methods in a new scenario.

**Weaknesses:**

- No effort has been made to fuse the proposed pipeline into a medical-image analysis specific methodological contribution. Why is for example the output temporally smoothed instead of using spatio-temporal consistency in higher dimensional networks? Why hasn't the semi-supervised paradigm be explored in more detail instead of only using a few biasing iterations with user input?
- A radical ablation study is clearly missing here. The task itself would imply that a deep network classifier is potentially an overkill. Bluntly: surgical parts are predominantly red, non-surgical parts anything and blue/green. How would a generic linear classifier on the image histograms perform here, or perceptual hashing with a linear classifier on top? Do we really need a labelled ground truth here? Can't simple heuristics perform at least as well? Assessing in-focus will even get rid of blurred frames and frames as discussed in the Appendix. There will be domain shift problems for the simple methods but same is true for the presented method.
- Writing, experimental setup and methodological proposals need to be improved and condensed.

**Detailed Comments:**

Overall the paper fits into the relevant problem of data preparation for downstream (learning) tasks. Above, I made several suggestions on how to improve this work and what aspects are missing in the current manuscript. A generic method to quickly cleanse data, making it also regulatory compliant would be very important. However, the presented paper does not provide such a tool; not only because neither code or data is provided to reproduce this work, also because simpler methods might be more effective for such data curation steps.

**Justification Of Rating:**

I have been working in this field for many years and published papers about these topics. I am advising regulatory decision makers and do active research in clinical environments. I am advocating open data access and reproducible research.

**Paper Type:**

validation/application paper

**Questions To Address In The Rebuttal:**

a) How would a simple histogram-based etc. classifier perform on this task?
b) how well does the trained network transfer to another clinical site (also compared to above)?
c) will the data be made publicly available after successful cleaning with the proposed method?


**Special Issue:**

no

---

> ### Author Response · Authors · 2020-03-26
> **We thank the reviewer for reading the manuscript and providing us with these constructive remarks.**
>
> In the weaknesses section, the reviewer wrote that “No effort has been made to fuse the proposed pipeline into a medical-image analysis specific methodological contribution”. While it is true we have not included a full end-to-end tool using these results in the paper for space reasons, we would contend that the path toward creating such a tool is relatively straightforward. Indeed, our end goal is to incorporate this into a full pipeline for surgical video analysis and, as the reviewer surmises, we have seen performance improvements by pre-filtering out-of-body segments. As these results mature, we will publish a more systems-focused report, but we felt these results stood on their own as a component for many surgical data science applications.
>
> With regard to the remark that “a deep network classifier is potentially an overkill”, a simpler classifier can handle this task to some extent, but it does not reach the same accuracy as our method. Given the sensitivity of privacy-related concerns, achieving very high performance is essential. As we explored the prior work, we quickly found that manually engineered features, thresholding and similar heuristic analysis always led to a significant number of edge cases that simply were not easy to handle. Our semi-supervised approach shows we can reap the substantial performance benefits of a DCNN with almost no up-front labeling effort.
>
> Addressing the rebuttal questions:
>
> 1) How would a simple histogram-based etc. classifier perform on this task?
>
> Response: In our preliminary studies, we implemented [Twinanda et al., 2014b] which used histogram-based scores to represent each video frame. These scores are used to empirically set thresholds on the RGB channels and classify non-relevant frames. The authors did not report the thresholds they found or the method performance. We implemented the method and found thresholds based on our train set. After evaluating these thresholds on our validation and test sets the results were significantly lower than what we show in our paper.
>
> Twinanda et al.:
> Valid / Test:
> Mean unweighted accuracy: 74.2% / 70.4%
> Precision: 28.6% / 30.5%
> Recall: 63% / 56.4%
>
> Ours:
> Valid / Test:
> Mean unweighted accuracy: 97.8% / 97.35%
> Precision: 78% / 83.5%
> Recall: 98.2% / 97%
>
> This comparison clearly shows that a simple histogram-based method is not able to generalize well to this problem, and doesn’t yield a robust classifier, especially compared to a deep neural network.
>
> We acknowledge that a different classifier using more sophisticated hand-engineered features  may perform better than the method proposed by Twinanda et al. However, there are few cases where such methods outperform DCNNs on such tasks, and, at a recall of 97%, achieving substantial improvements without sacrificing precision seems quite challenging. Conversely, a drop of even a few percent in precision would potentially double the chance of missed frames, thus greatly increasing the downstream risk for an end-user and lowering the acceptability of the method. For this reason, we ceased to seriously pursue other approaches and instead focused on refining the performance of a DCNN.
>
> 2) how well does the trained network transfer to another clinical site (also compared to above)?
>
> Response: Our dataset was curated from several different medical centers, and thus span a variety of recording systems. Calculating the mean unweighted accuracy of each center separately in the validation and test set yields the following mean and std values:
> Validation mean=0.975 and std=0.0146
> Test mean=0.955 and std=0.0283
> Thus showing that our method generalized well across different medical centers and recording systems.
> This is significantly better than the results of Twinanda et al.:
> Validation mean=0.676 and std=0.096
> Test mean=0.685 and std=0.0712
>
> 3) will the data be made publicly available after successful cleaning with the proposed method?
>
> Response: The videos that support the findings of this study unfortunately contain restricted data that, due to IRB and licensing limitations, we are unable to make publicly available. We hope this work will however accelerate future sharing of surgical videos by providing a simple and reliable anonymization with a method like the one described in this study.

---

> > ### Comment · AnonReviewer1 · 2020-03-29
> > **very convincing reply, I would change my recommendation to 'accept'**
> >
> > Thank you for providing these insights! Would it be possible to add them to the paper (or an Appendix) please?
> > re data: I suspected such constraints, but it's a bit strange to have a paper that claims to be useful for exactly the privacy part of data sharing, which then does not follow up by providing the resulting, perfectly anonymized data.  Licensing issues can easily be resolved by adding a (restrictive) license to the data.

---

> > > ### Author Response · Authors · 2020-03-31
> > > **Yes, we will add these results to the revised version.**
> > >
> > > Regarding data release, we are working on an initiative to introduce a large surgical video dataset that will allow access in a privacy-preserving way. The current work, of automatically anonymizing large surgical datasets with almost no labeling effort, started as part of this initiative. We hope to release the dataset soon, as a conference challenge or a different publication.

---

> > > > ### Comment · AnonReviewer1 · 2020-03-31
> > > > **great, looking forward to this!**
> > > >
> > > > ...

---

> > > > ### Comment · AnonReviewer3 · 2020-04-01
> > > > **Looking forward to it as well <EOM>**

---

### Official Review · AnonReviewer4 · 2020-03-13
**An interesting work on the detection of non-relevant surgery segments but of limited novelty.**

**Rating:** 2
**Confidence:** 5

**Summary:**

The paper presents a deep learning-based framework for the classification of surgery-related and non-relevant segments in surgical videos. For this purpose, the ResNet-18 model has been employed and a weakly-supervised approach combined with iterative semi-supervised learning has been proposed to annotate the training dataset. The performance of the proposed method has been evaluated on laparoscopic cholecystectomy videos.

**Strengths:**

This is an interesting work which fits well to the scopes of the conference. The paper is well written and easy to follow. The method seems theoretically sound and the references adequate. The performance evaluation had been based on the analysis of surgical video sequences.

**Weaknesses:**

The technical novelty of the proposed method is limited. A state-of-the-art network has been used for data classification and the main contribution is the different approaches to generate the training dataset. Also, the clinical motivation is not very strong.

**Justification Of Rating:**

As stated at the "Weaknesses" section, the presented work is of limited technical novelty and the clinical motivation is not very strong. I believe this paper is not ready to be accepted for publication at this conference.

**Paper Type:**

both

**Questions To Address In The Rebuttal:**

•	The title does not reflect the proposed work very well as “anonymisation” is one of the applications and should not be presented as the only aim of the method.
•	The clinical motivation is not very strong. How many sequences actually contain information which needs to be automatically removed for anonymisation purposes?
•	In Section 2.1, it is mentioned that the training dataset is split into 6 subsets but the aim of this is only explained in Section 3.
•	It would be interesting to show the performance of the supervised classification.
•	For a more robust validation, the method should be compared to state-of-the-art methods for example,  [Munzer et al., 2013] and [Twinanda et al., 2014b].
•	The technical novelty of the proposed method is limited as it mainly employs a state-of-the-art network for data classification and proposes different semi-supervised approaches to train the classification model.
•	The figures are very small, making it difficult to observe the details, particularly for Fig. 7-8.


**Special Issue:**

no

---

> ### Author Response · Authors · 2020-03-26
> **We would like to thank the reviewer for the valuable comments on the manuscript. We appreciate the opportunity to clarify our research and made a few changes in the revised manuscript based on the proposed suggestions.**
>
> In the weaknesses section the reviewer wrote that “The technical novelty of the proposed method is limited” and that “the clinical motivation is not very strong”.
>
> While we agree our technical approach is straightforward, in fact both the problem setting and solution were driven by an urgent need to develop scalable, clinically deployed video-based surgical data science tools. As we moved our solutions toward deployment, privacy concerns emerged as a key issue that potentially limited interest in sharing data for quality improvement and educational purposes. Further, we needed a light-weight solution that could be readily adapted as necessary which meant it must be adaptable to other procedures without requiring detailed annotation. As a side-effect, we also gain a method to scrub the data set of irrelevant video which leads to cleaner data for training. Thus, both the problem setting and solution emerged from a specific clinical need, and address both technical and clinical constraints.
>
> 1) The title does not reflect the proposed work very well as “anonymisation” is one of the applications and should not be presented as the only aim of the method.
>
> Response: We changed the title to:
> “Accurate Detection of Out of Body Segments in Surgical Video using Semi-Supervised Learning”
>
> 2) The clinical motivation is not very strong. How many sequences actually contain information which needs to be automatically removed for anonymisation purposes?
>
> Response: As noted above, due to privacy and regulatory concerns it is effectively necessary to remove all out-of-body video segments from any submitted video as any out-of-body segment could contain a face, or (e.g. in Vaginal Hysterectomy) inappropriate image.
>
> To better make this point, we have modified Table 1 to show the number of out-of-body segments (this includes all segments, not only those which contain information that specifically relate to anonymization). The validation set has 64 and the test set has 79 such segments. Each set has 20 videos, thus on average, each video contains 3-4 non-relevant segments (not incl. start/end segments) that need to be removed, for anonymization and curation purposes.
>
> 3) In Section 2.1, it is mentioned that the training dataset is split into 6 subsets but the aim of this is only explained in Section 3.
>
> Response: We changed the first paragraph of Section 2.1 adding the following:
> “To explore different training setups, in which (1) the same train set and (2) a different train set of 100 videos are used for training, we randomly divided the remaining 600 videos into six train subsets, from which one subset is defined as the initial baseline train set in our experiments.”
>
> 4) It would be interesting to show the performance of the supervised classification.
>
> Response: An actual supervised training is not possible since the train set is not fully annotated. Training on all of the available labels still uses many noisy labels that our method is trying to fix by iteratively cleaning the train set. In our experiments training on the initial set of noisy labels is considered as the first training iteration throughout the different experiments.
>
> 5) For a more robust validation, the method should be compared to state-of-the-art methods
>
> Response: Comparing our method results to the more recent study of Twinanda et al. is not straightforward, as the authors did not report their method performance or the threshold values. However, in order to achieve some comparison we followed their method and calculated RGB histograms for our train set, these are then used to select the best thresholds for classification. Applying these thresholds on the validation and test sets yield significantly lower results compared to our method.
>
> Twinanda et al.:
> Valid / Test:
> Mean unweighted accuracy: 74.2% / 70.4%
> Precision: 28.6% / 30.5%
> Recall: 63% / 56.4%
>
> Ours:
> Valid / Test:
> Mean unweighted accuracy: 97.8% / 97.35%
> Precision: 78% / 83.5%
> Recall: 98.2% / 97%
>
> 6) The technical novelty of the proposed method is limited as it mainly employs a state-of-the-art network for data classification and proposes different semi-supervised approaches to train the classification model.
>
> Response: As noted above, our motivation is to provide a robust, easily retargetable method able to detect out-of-body and non-relevant segments in high accuracy for privacy and curation reasons. What is technically novel is that a relatively simple iterative training leads to a network with very high performance with almost no labeling effort. Further, as noted above, the performance of manually engineered features or heuristic-based thresholds is markedly inferior (e.g. Twinanda et al.) and would not be sufficient for our purposes. Indeed, we see the simplicity of the method as a key contribution as these results overcome a key barrier for surgical data science that is easily implemented by other groups.

---

> > ### Author Response · Authors · 2020-03-26
> > **Questions To Address In The Rebuttal - continue:**
> >
> > 7) The figures are very small, making it difficult to observe the details, particularly for Fig. 7-8.
> >
> > Response: We increased the size of all figures.

---

### Official Review · AnonReviewer3 · 2020-03-14
**Office review**

**Rating:** 3
**Confidence:** 5
**Recommendation:** Poster

**Summary:**

This paper proposes a semi-supervised learning method for surgical video anonymization. The proposed method is first trained on a roughly labeled dataset containing unlabeled part as noise; then the training set keeps changing during iterations in training process. The method is interesting and well organized, especially for the visualized results in experiments, while more details could be given to make the work more solid.

**Strengths:**

Surgical video anonymization is a very interesting and significant research topic and it is worth being discussed as privacy become increasingly important in research and products. This paper introduces a deep learning model to solve this problem, and proposed a semi-supervised learning way to avoid frame level labeling. The paper is well organized and presented.

**Weaknesses:**

1) The title is a bit confusing. Anonymization does not exactly describe the thing the paper and the proposed method focus. What the paper is doing seems like to denoise surgical videos in a temporal dimension, removing irrelevant frames in surgical videos.
2) The usage of semi-supervised is also confusing. The training data is only labeled for the before-starting and after ending parts of the video, while considering the irrelevant frames inside the surgery as relevant. It is more close to handling noisy datasets, but kind of different from general semi-supervised learning.
3) The validation set should be fully labeled. Only in this way, could the iteratively changing the training dataset be evaluated  with certain metrics.

**Justification Of Rating:**

This paper has good overall quality and a weak accept is appropriate. I would suggest a high rating if the presentation, wording is more accurate and the experiment design is better designed and taken.

**Paper Type:**

validation/application paper

**Special Issue:**

no

---

> ### Author Response · Authors · 2020-03-26
> **We thank the reviewer for the interest in our work and the helpful comments.**
>
> To address the weaknesses raised we made the following changes in the revised version:
>
> 1) The title is a bit confusing. Anonymization does not exactly describe the thing the paper and the proposed method focus. What the paper is doing seems like to denoise surgical videos in a temporal dimension, removing irrelevant frames in surgical videos.
>
> Response: The anonymization was referring to the main application of out-of-body segments detection. We accepted the comment and changed the title to better describe the paper focus and the proposed method:
> “Accurate Detection of Out of Body Segments in Surgical Video using Semi-Supervised Learning”
>
> 2) The usage of semi-supervised is also confusing. The training data is only labeled for the before-starting and after ending parts of the video, while considering the irrelevant frames inside the surgery as relevant. It is more close to handling noisy datasets, but kind of different from general semi-supervised learning.
>
> Response: In order to emphasize the fact that the initial dataset is noisy we adjusted the following sentence in the abstract to make sure it is clear for the reader:
>
> “Since annotating surgical videos with per-second relevancy labeling is a tedious task, our proposed framework initiates the learning process from a weakly labeled noisy dataset and iteratively applies Semi-Supervised Learning (SSL) to re-annotate the training data samples.”
>
> In addition, we changed the last paragraph of the introduction section to highlight this fact even further:
>
> “Labeling the start-end task leads to a weakly labeled noisy dataset, consists of a large number of positive segments and a noisy set of negative segments which include positive seconds falsely labeled as negative. Our method utilizes this noisy set of labeled data as the basis for training a model for the full task of non-relevant detection. This is done by iteratively training a model with incomplete supervision.”
>
> 3) The validation set should be fully labeled. Only in this way, could the iteratively changing the training dataset be evaluated  with certain metrics.
>
> Response: The validation and test sets were indeed fully labeled manually, as described at the end of the second paragraph in Section 2.1: “In contrast, the evaluation sets were manually annotated with full coverage of non-relevant labeling throughout the entire surgical video (Table 1).”
> To make it clearer we changed the phrase “evaluation sets” to  “validation and test sets”.

---

### Meta-Review · Area_Chair1 · 2020-04-06
**MetaReview of Paper157 by AreaChair1**

**Rating:** 3
**Recommendation For Accepted Papers:** Poster

**Metareview:**

The reviewers agree that the application presented in this manuscript is of interest but alos point to a lack of methodological contributions and to some shortcoming in the evaluation of teh method. The authors have nonethless provided interesting responses in their rebuttal and the paper might lead to interesting discussion at the conference.

**Paper Type:**

validation/application paper

**Special Issue:**

no

---

### Decision · Program_Chairs · 2020-04-11

Accept